# Predictors of Single Bronchodilation Treatment Response for COPD: An Observational Study with the Trace Database Cohort

**DOI:** 10.3390/jcm10081708

**Published:** 2021-04-15

**Authors:** Laura Carrasco Hernández, Candela Caballero Eraso, Borja Ruiz-Duque, María Abad Arranz, Eduardo Márquez Martín, Carmen Calero Acuña, Jose Luis Lopez-Campos

**Affiliations:** 1Unidad-Quirúrgica de Enfermedades Respiratorias, Instituto de Biomedicina de Sevilla (IBiS), Hospital Universitario Virgen del Rocio/Universidad de Sevilla, 41013 Sevilla, Spain; lauracarrascohdez@gmail.com (L.C.H.); ccaballero-ibis@us.es (C.C.E.); borja_994@hotmail.com (B.R.-D.); drabadarranz@gmail.com (M.A.A.); eduardo.marquez.sspa@juntadeandalucia.es (E.M.M.); ccalero-ibis@us.es (C.C.A.); 2CIBER de Enfermedades Respiratorias (CIBERES), Instituto de Salud Carlos III, 28029 Madrid, Spain

**Keywords:** COPD, long-acting bronchodilators, clinical response, pharmacological

## Abstract

Chronic obstructive pulmonary disease (COPD) patients constitute a heterogeneous population in terms of treatment response. Our objective was to identify possible predictive factors of response to treatment with single bronchodilation monotherapy in patients diagnosed with COPD. The Time-based Register and Analysis of COPD Endpoints (TRACE; clinicaltrials.gov NCT03485690) is a prospective cohort of COPD patients who have been attending annual visits since 2012. Patients who were kept on a single bronchodilator during the first year of follow-up were selected. The responders were defined according to all of the following variables: any improvement in morning post-dose forced expiratory volume in 1 s or deterioration <100 mL, no change or improvement in dyspnea score, and no occurrence of exacerbations. Significant and plausible variables were analyzed using a proportional hazard Cox regression for single bronchodilator responders. We analyzed 764 cases, of whom 128 (16.8%) were receiving monotherapy with one bronchodilator. Of these, 85 patients (66.4%) were responders. Factors affecting responder status were: female gender (hazard ratio (HR) 0.276; 95% confidence interval (CI) 0.089–0.858), dyslipidemia (HR 0.436; 95%CI 0.202–0.939), not performing regular exercise (HR 0.523; 95%CI 0.254–1.076), active smoking (HR 0.413; 95%CI 0.186–0.920), and treatment adherence (HR 2.527; 95%CI 1.271–5.027). The factors associated with a single bronchodilation response are mainly non-pharmacological interventions and comorbidities.

## 1. Introduction

Recently, our understanding of the pathophysiology and biological pathways of chronic obstructive pulmonary disease (COPD) has advanced, and the number of therapeutic options has increased. As a result, different approaches have been proposed over the last few years for the pharmacological management of this condition that aim at promoting patient-directed approaches [1]. The popular Global Initiative for Obstructive Lung Disease (GOLD) document suggests differential pharmacotherapeutic approaches to the available therapies depending on the symptom burden and exacerbation risk [2]. In particular, the GOLD document recommends the use of a single long-acting bronchodilator (LABD) to initiate maintenance therapy and the combination of two LABDs for cases with more severe dyspnea and exacerbations, together with the use of inhaled corticosteroids (ICSs) for frequently exacerbating patients combined with a peripheral blood eosinophilic count.

However, COPD patients are a heterogeneous population group in terms of treatment response [3,4,5]. Interestingly, a large number of patients do not show the expected improvement in terms of symptoms and number of exacerbations, which significantly reduces their quality of life and worsens the clinical presentation [6]. Accordingly, each pharmacological treatment regimen should be tailored to and guided by the severity of symptoms, risk of exacerbation, side effects, comorbidities, and, most crucially, by the patient’s response to the treatment. In this scenario, although the efficacy of double bronchodilation combined with an ICS is well supported by clinical trials, currently there is a lack of knowledge about how to identify patients who do not require any more therapy than the single bronchodilation monotherapy treatment. 

The Time-based Register and Analysis of COPD Endpoints (TRACE) study is a prospective ongoing cohort of patients with COPD [7]. Here, we aimed to study the demographic, sociodemographic, and clinical risk factors that are predictors of the patient’s response to single bronchodilation monotherapy treatment in COPD patients based on the prospective TRACE cohort population. The objective was to identify possible predictive factors of response to treatment with single bronchodilation monotherapy in patients diagnosed with COPD, and to identify which characteristics presented by these patients at the moment they initiate LABD monotherapy treatment may predict a positive clinical response. The results of our study will potentially help clinicians in their daily clinical decision-making.

## 2. Materials and Methods

The methodology of TRACE has previously been published [7,8]. Briefly, TRACE (clinicaltrials.gov NCT03485690) is a prospective observational study of non-interventionist cohorts from a single center. The sample consists only of COPD patients, identified according to the current diagnostic criteria [9]. The protocol does not prespecify any exclusion criteria, except for the complete reversibility of lung function tests during follow-up.

Patient inclusion began in January 2012. After case identification, patients are followed up on prospectively at annual visits sine die until they die or are lost for follow-up. All the subjects receive their prescribed medications and therapeutic interventions throughout the study and any changes in medication are ordered by the physician in charge, according to the clinical condition of the patient. During annual visits, clinical, functional, radiological, and analytical information is recorded using a standardized questionnaire for all visits. 

The variables collected are: sociodemographic (gender, age), tobacco history, comorbidities (Charlson index [8]), clinical presentation with stable situation during the previous year (including evaluation of dyspnea, cough and sputum production, sputum color if present, and self-referred wheezing), exacerbations and hospitalizations in the previous year, current pharmacological and non-pharmacological treatment and ancillary tests, including at least simple chest radiology, pre- and post-bronchodilator spirometry and analytical results (blood eosinophils, alpha1-antitrypsin, C-reactive protein, and total immunoglobulin E). Exacerbations are collected by enquiring the patients. Additionally, this information is matched with that in the medical record to avoid recall bias.

### Statistical Analysis and Modeling

The present analysis includes the progression of TRACE patients during the first year of follow-up including the first two visits (basal and first-year control). The analysis population is defined as the population of patients who completed the follow-up visit. Qualitative variables are described by frequency and percentages based on the size of the complete data, while the quantitative variables are described by the mean and standard deviation. 

Patients receiving an LABD as monotherapy (once or twice daily) in the first visit were selected and divided into two groups according to the clinical response to this LABD (responders and non-responders). The response criteria were described as a composite result including dyspnea increase, exacerbations, and lung function deterioration during the one-year follow-up. Accordingly, the responders were defined as those who presented all of the following variables during the follow-up: deterioration < 100 mL or any improvement in morning post-dose forced expiratory volume in 1 s (FEV_1_), no change or improvement in dyspnea score according to the modified Medical Research Council scale [9], and no occurrence of exacerbations. Patients not fulfilling any one of these three criteria were considered non-responders.

The statistical program PSPP (GNU pspp 1.2.0-g0fb4db) was used for calculation and transformation of variables. The statistical analysis was conducted using RStudio v. 1.1.456. The alpha error was set to 0.05. To describe the population better, an analysis of the LABD users was included and compared with the rest of the cohort. To identify the factors related to the response to LABD in monotherapy, a bivariate analysis was initially performed between LABD monotherapy responders and non-responders. For the qualitative variables, comparisons were made using the chi-square test for independent groups. If the observed frequency was lower than expected, Fisher’s exact test was performed. For the quantitative variables, the unpaired t-test was used with the evaluation of homoscedasticity using the Levene test. Significant and plausible variables were entered into a proportional hazard Cox regression for LABD responders to assess the independence of the effects. The results are expressed as a hazard ratio (HR) with 95% confidence intervals (CIs). The goodness of fit of the model was determined using the Wald statistic, in addition to the level of agreement of the model and the standard error.

## 3. Results

From the 962 patients in the TRACE cohort database, we selected 764 cases with at least one year’s follow-up, of whom 128 (16.8%) were receiving monotherapy with one LABD (25 cases with a long-acting beta2-agonist and 102 cases with a long-acting muscarinic antagonist). Of these, 85 patients (66.4%) were responders and 43 (33.6%) were non-responders. Patients not under LABD monotherapy were receiving in 121 cases (15.8%) two LABDs, in 11 cases (1.4%) a single ICS, in 101 cases (13.2%) a combination of ICSs and LABDs, in 357 cases (46.7%) triple therapies, and in 46 cases (6.0%) no maintenance therapy at all (only rescue medication). 

The description of the study population showing the differences between LABD monotherapy cases vs. those receiving other therapies is summarized in Table 1. As expected, LABD-only users presented clinical differences from the rest of the cohort, being younger, having less hypertension, less severe lung function impairment, fewer symptoms as measured by the Medical Research Council (MRC) dyspnea scale, less cough and expectoration, a smaller number of exacerbations, doing exercise more frequently, and being less likely to receive the influenza vaccine and oxygen therapy. These results suggest that patients in the LABD treatment group had milder disease and fewer symptoms at baseline than patients in the other therapies group.

The final multivariate Cox regression model is shown in Figure 1. The model obtained allowed us to differentiate those clinical variables that influence or could predict the clinical response of the patients that have initiated the treatment with monotherapy. In particular, treatment adherence increased the likelihood of being a responder, whereas female gender, dyslipidemia, being an active smoker, body mass index, and previous exacerbations decreased this possibility. The exercise variable shows an association with response to therapy, suggesting that it could have a protective effect, and patients who practice exercise would therefore show better progression compared to those who do not, although these results did not reach the pre-specified significance threshold limit.

## 4. Discussion

The results of clinical trials revealed relevant information on the efficacy and safety of inhaled drugs for the treatment of COPD. These trials show the average responses of a cohort of patients analyzed globally. However, it is known that the response described does not correspond with a response of similar magnitude at the patient level [10,11,12]. In this context, the clinical trials tell us that a patient is more or less likely to respond to a certain therapy, but without being able to give us valid information about the effect on a specific patient. Consequently, real-life studies are needed to analyze the effectiveness of treatments at the patient level [13]. The present analysis identifies the variables associated with an adequate therapeutic response to a single bronchodilator in patients with COPD. According to the results, on the one hand, treatment adherence is the main factor associated with a positive response to an LABD; on the other, gender, cardiovascular risk factors like dyslipidemia, not performing daily exercise, and being an active smoker decrease the likelihood of responding to an LABD.

TRACE is a prospective observational study whose objectives include the assessment of the therapeutic response at the patient level [7]. The present analysis has some strengths, including the real-life description of a prospective cohort with a large sample size and the capacity to identify individual responses based on clinical grounds. Notably, some methodological considerations must be taken into account in the study to interpret the results correctly. Firstly, although the starting sample size is large, the number of patients receiving monotherapy is more limited, which results in decreased statistical power and some potential associations not reaching statistical significance. The advancement of new treatments for COPD that provide greater bronchodilator and anti-inflammatory power, as well as the combination of several drugs in a single inhalation device, has led to single therapies being used less often [14,15,16,17]. In reality, this attitude towards stepping-up therapy has been widely adopted, although it may not be the most suitable approach, as we point out in our work. Secondly, TRACE is a cohort study that collects information from patients with COPD at a hospital consultation specifically dedicated to COPD. These patients therefore have a greater degree of severity than if a primary care cohort or a cohort with milder patients had been selected. Consequently, it would be necessary to validate these results in cohorts with other patient profiles. Thirdly, according to the protocol, the variables collected in TRACE are those that can be obtained from a routine clinical visit with accessible diagnostic methods at a conventional healthcare unit. This means that TRACE does not include extraordinary diagnostic measures beyond routine clinical practice so that the results can be assessed at the usual clinic. Accordingly, the database does not contain information from more specific diagnostic techniques like impulse oscillometry or peripheral muscle strength evaluation. However, by using other, more precise, diagnostic tools or questionnaires which are more sensitive to change, it may be possible to obtain different results. Interestingly, due to the objectives of the TRACE cohort, lung volume or diffusing capacity was not recorded in all patients. This measurement would have been interesting to describe the impact of single LABD treatment on lung hyperinflation in those responding to therapy vs. those not. Finally, the database does not record the dose regimen. Therefore, it is not possible to explore the potential impact of 12 vs. 24 h in the efficacy of single bronchodilation.

As regards the sociodemographic variables, we found differences in the response to treatment with respect to the gender of the patients, indicating that the patient’s gender is a factor associated with the response to treatment with LABD monotherapy. Numerous studies have highlighted the differences in the clinical presentation of COPD between men and women, with women being more symptomatic, although this does not have a clear impact on survival [18,19]. However, the therapeutic response has been consistently reported to be similar in clinical trials and no observational study has shown a different response beyond a higher probability of bronchial hyperresponsiveness and the potential use of ICSs [20]. Our findings need to be taken with caution due to the increased number of males in our cohort. Now that the epidemic of COPD in women is beginning to emerge [21], in the coming years we will be able to obtain enough data to come to a more balanced view on the impact of treatments according to gender in COPD. In contrast, age does not seem to have a significant effect on response to treatment. This jeopardizes the traditional idea that COPD unquestionably progresses with age, in other words, that the older the lung function, the weaker the response to single bronchodilation [22]. In line with other previous authors, our data contradict this view of the disease [23].

Among the relevant findings of our analysis are that the variables that the disease management recommendation documents typically rely on (symptoms and exacerbations) were not as significant as expected. In contrast, with patients from a real-life clinical context, other variables such as adherence to treatment, active smoking, or exercise did have a predictive role in therapeutic response. The importance of adherence to treatment perhaps does not require too much emphasis, since we are currently well aware of the importance of maintaining good adherence to treatment and the possible clinical impact that it entails [24]. However, this result should make us reflect on the importance of getting the patient to adhere well to the therapeutic regimen before proceeding with the escalation of treatment. The relationship between active smoking and the response to an LABD, however, is a more novel finding. Until now, various studies have pointed to the possibility of an interaction between active smoking and the therapeutic response to the use of ICSs [25]. However, the information available on the relationship between tobacco and the therapeutic response to an LABD is less consistent. Our data clearly indicate that the active smoking population is a population at risk of having a lower response to an LABD. Although these data must be confirmed in another cohort with a larger number of patients with single bronchodilation, the association is worth highlighting, since it stresses the essential role of smoking cessation as part of a complete therapeutic strategy in COPD patients [26]. Finally, although, strictly speaking, the relationship did not reach statistical significance, we did find an association close to significance between the response to an LABD and performing exercise. It is well known that physical exercise has a number of clinical benefits on the impact of the disease at various levels. Therefore, it is expected that patients who perform daily exercise are more aware of the importance of controlling the disease and, therefore, less frequently report the need for an escalation of treatment. Furthermore, the combination of physical exercise and bronchodilator treatment has been shown to be a convenient therapeutic approach for patients with COPD. In fact, in this context, one question remains unanswered, which is whether a second LABD or a respiratory rehabilitation program would be the best strategy for a symptomatic patient who is taking a LABD [27]. Although we cannot answer this question with the current design, it is true that the addition of a respiratory rehabilitation program has both benefits and drawbacks compared to dual bronchodilator therapy. On the one hand, it is true that rehabilitation achieves clinical benefits that go beyond the clinical and functional improvement of bronchodilators [28]. On the other hand, it is also true that not all patients respond equally well to rehabilitation, that it requires a series of resources that are not always available, and that its beneficial effects are transitory and do not last beyond the training program [29]. Therefore, this decision constitutes a key step in personalizing treatment. Our data support the role of exercise as the main component of the holistic therapeutic approach to COPD patients.

In addition, the role of the proper management of comorbidities in the COPD patient is another interesting point for analysis [27]. Although not always statistically significant, four comorbidities remain in the multivariate model: arterial hypertension, dyslipidemia, body mass index, and sleep apnea–hypopnea syndrome. With the data provided by our work, these associations are difficult to put into context, since the relationship of COPD with its comorbidities, especially cardiovascular comorbidities, is complex. The use of statins in COPD has also been explored. Although it seems that their role in the treatment of COPD is still questioned [30], various studies have pointed to a beneficial effect [31,32]. If this finding is confirmed, it could explain the reduction in the risk of non-response in this group of patients. In addition, the relationship of the body mass index with the progression of COPD has also been revealed in numerous studies [33]. Our data indicate, in line with the literature, that having a healthier body mass index is associated with not having to escalate bronchodilator medication. Of note, obesity has also been associated with bronchial hyperresponsiveness [34,35] and could explain a less relevant response in these cases. These findings are in line with the available literature and emphasize the need to establish a holistic therapeutic approach that is not exclusively focused on the prescription of inhaled medication. On the contrary, with COPD, non-pharmacological treatments have shown stronger benefits than pharmacological ones.

## 5. Conclusions

In summary, the present work explores the variables associated with a positive therapeutic response to single bronchodilator treatment. Our analysis reports that factors such as active smoking, therapeutic adherence, non-pharmacological treatment, and some comorbidities may have a relationship with the effectiveness of single bronchodilation. In particular, it highlights the potential role of treatment adherence and inhalation technique in the evaluation of therapeutic response. Therefore, before recommending an escalation of therapy, it would be advisable to employ various aspects of the therapeutic approach in order to follow a holistic strategy in the treatment of COPD.

## Figures and Tables

**Figure 1 jcm-10-01708-f001:**
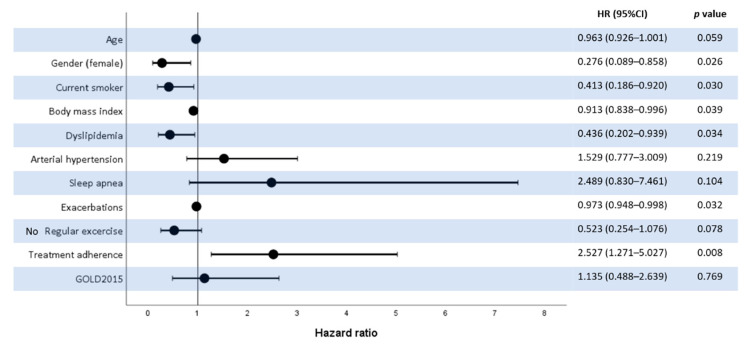
Forest plot showing the results of the Cox proportional hazard regression of factors associated with being a responder to single bronchodilation as a treatment for COPD. Note: HR—hazard ratio; CI—confidence interval.

**Table 1 jcm-10-01708-t001:** Description of chronic obstructive pulmonary disease (COPD) cases with long-acting bronchodilator (LABD) monotherapy vs. other therapies.

Factors	Total (*n* = 764)	LABD Monotherapy (*n* = 128)	Other Treatments (*n* = 636)	*p*-Value *
Age (years)	69 (62.76)	66 (62.73)	70 (61.77)	0.017
Gender (male)	6,590,020 (86.3)	108 (84.4%)	551 (86.6%)	0.498
Comorbidities (Charlson)	2.26 (1.6)	2.13 (1.4)	2.28 (1.6)	0.309
Hypertension	389 (51.0%)	53 (41.7%)	336 (52.9%)	0.028
FEV_1_ (mL)	1320 (720)	1700 (757.5)	1260 (617.5)	<0.001
GOLD 1	71 (9.3%)	25 (19.7%)	46 (7.2%)	<0.001
GOLD 2	411 (53.8%)	89 (69.5%)	322 (50.6%)	<0.001
GOLD 3	227 (29.7%)	13 (10.2%)	214 (33.6%)	<0.001
GOLD 4	55 (7.2%)	1 (0.8%)	54 (8.5%)	0.002
MRC 0	144 (18.9%)	49 (38.6%)	95 (15.0%)	<0.001
MRC 1	393 (51.4%)	61 (47.2%)	332 (52.2%)	0.384
MRC 2	150 (19.7%)	17 (13.4%)	133 (20.9%)	0.047
MRC 3	55 (7.2%)	1 (0.8%)	54 (8.5%)	0.002
MRC 4	18 (2.4%)	0 (0.0%)	18 (2.8%)	0.054
Cough	451 (59.2%)	63 (49.6%)	388 (61.1%)	0.021
Expectoration	407 (53.4%)	57 (44.9%)	350 (55.1%)	0.044
Exacerbations during the previous 12 months	0.9 (1.1)	0.6 (0.9)	0.9 (1.1)	<0.001
Hospital admission during the previous 12 months	0.1 (0.4)	0.08 (0.2)	0.1 (0.4)	0.124
Perform daily exercise	212 (27.8%)	46 (36.2%)	166 (26.1%)	0.027
Received flu vaccine	526 (69.0%)	75 (59.1%)	451 (71.0%)	0.011
On oxygen therapy	83 (10.9%)	2 (1.6%)	81 (12.8%)	<0.001

Note: FEV_1_—forced expiratory volume in 1 s; GOLD—Global Initiative for Obstructive Lung Disease; MRC—Medical Research Council. Results expressed as mean (standard deviation) or as absolute frequencies (with relative frequencies in parentheses). * Calculated by chi-square test or unpaired t-test depending on the nature of the variable.

## Data Availability

Data are available for researchers by demand upon reasonable request.

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
