# Peer review of "Predictors of Single Bronchodilation Treatment Response for COPD: An Observational Study with the Trace Database Cohort"

_jcm, 2021, doi:10.3390/jcm10081708_

Round 1

Reviewer 1 Report

I have reviewed this interesting article on the determinants of the therapeutic response to a single bronchodilator in the patient with COPD. The authors analyzed the TRACE cohort and identified some response markers in relation to non-pharmacological interventions and comorbidities. The work is interesting and well executed. Some comments are as follows:

1. I wish they had separated the LAMAs from the LABAs, which would have given some difference between the two therapeutic families.

2. I have doubts about the external validity of the results, as they are patients obtained from specialized pulmonology consultations.

3. It is possible that if the authors had included other functional variables such as impulse oscillometry or measurement of volumes, diffusion or peripheral muscle strength, they could have had a more complete analysis.

4. How exacerbations were collected at each visit?

5. In the conclusions, the authors should emphasize the role of adherence and inhalation technique in the escalation strategy they want to explore.

Author Response

COMMENT. I have reviewed this interesting article on the determinants of the therapeutic response to a single bronchodilator in the patient with COPD. The authors analyzed the TRACE cohort and identified some response markers in relation to non-pharmacological interventions and comorbidities. The work is interesting and well executed. Some comments are as follows:

ANSWER: we would like to thank the reviewer for the thorough evaluation of out manuscript. We will now answer each point below.

COMMENT 1. I wish they had separated the LAMAs from the LABAs, which would have given some difference between the two therapeutic families.

ANSWER: We agree with this comment. We initially thought about dividing the two families of bronchodilators. However, since LAMAs are more frequently used, we lost statistical power with the LABAs and the results were not consistent. Additionally, in real-life the interesting question is the response to one long-acting bronchodilator as the GOLD recommends. So, we finally analyzed them together.

COMMENT 2. I have doubts about the external validity of the results, as they are patients obtained from specialized pulmonology consultations.

ANSWER: The reviewer is correct. These patients are all from a specialized COPD-dedicated clinic. As the reviewer indicates, although there are patients of different degrees of severity, this is a selected population and therefore somehow biased. This is commented in the discussion from line 172 on.

COMMENT 3. It is possible that if the authors had included other functional variables such as impulse oscillometry or measurement of volumes, diffusion or peripheral muscle strength, they could have had a more complete analysis.

ANSWER: This is a very interesting comment. The advent of new diagnostic devices for the evaluation of respiratory physiology implies an opportunity for a better characterization of the patients. However, one of the key aspects of TRACE is to be able to apply its findings for clinical decision in daily clinical practice. That is why TRACE does not use diagnostic tools that are not available in a routine medical visit for respiratory patients. Therefore, the TRACE cohort does not collect data from these novel diagnostic devices. This is commented in the discussion from line 176 on.

COMMENT 4. How exacerbations were collected at each visit?

ANSWER: This is an important comment we forgot to mention in our first version of the manuscript. Exacerbations were collected by enquiring the patients. Additionally, this information was matched with that in the medical record to avoid the recall bias.

COMMENT 5. In the conclusions, the authors should emphasize the role of adherence and inhalation technique in the escalation strategy they want to explore.

ANSWER: We have added a comment in the conclusions, as per the suggestion.

Reviewer 2 Report

The study entitled “Predictors of single bronchodilation treatment response for COPD: an observational study with the TRACE database cohort” is a prospective cohort study in Spain. The  aim study was to valuate the demographic, sociodemographic and clinical risk factors which are predictors of the patient’s response to single bronchodilation monotherapy treatment in COPD.

  • Has been performed the acute bronchodilatation test?
  • What kind of LABA and LAMA have been used? Drugs once a daily or twice a daily? Should be interesting to report this point identifying how many pts used once a daily and how many the twice daily. Could this point influence the responders pts?
  • Did you have data on lung hyperinflation, IC or Sawr? This point could be important in order to understand the role of a single LABD use in studied COPD population where 89.2% were GOLD 1 & 2, so mild or moderate pts! Why a so good effect? Different evidences underline as a single LABD well work on lung deflation and consequently on dyspnoae (your data reported MRC from 0 to 2 = 99.2% of COPD pts with only a single LABD); see 1) Comparison of the acute effect of tiotropium versus a combination therapy with single inhaler budesonide/formoterol on the degree of resting pulmonary hyperinflation. Respir Med. 2006 Jul;100(7):1277-81. 2)Assessment of acute bronchodilator effects from specific airway resistance changes in stable COPD patients. Respir Physiol Neurobiol. 2014 Jun 15;197:36-45
  • Furthermore, the bronchodilatation and lung deflation working on dyspnoea and this could have influence also on the treatment satisfaction as well as on the adherence (see Satisfaction with chronic obstructive pulmonary disease treatment: results from a multicenter, observational study. Ther Adv Respir Dis. 2019 Jan-Dec;13:1753466619888128).
  • In general the Discussion is interesting but must be shortened for every point and implemented with a brief point relating to the two previous suggestions.

Author Response

The study entitled “Predictors of single bronchodilation treatment response for COPD: an observational study with the TRACE database cohort” is a prospective cohort study in Spain. The  aim study was to valuate the demographic, sociodemographic and clinical risk factors which are predictors of the patient’s response to single bronchodilation monotherapy treatment in COPD.

COMMENT. Has been performed the acute bronchodilatation test?

ANSWER: Yes, all spirometric values analyzed were post-bronchodilator.

COMMENT. What kind of LABA and LAMA have been used? Drugs once a daily or twice a daily? Should be interesting to report this point identifying how many pts used once a daily and how many the twice daily. Could this point influence the responders pts?

ANSWER: In the current analysis there is no differentiation between 12 or 24 h long-acting bronchodilators. Unfortunately, the TRACE cohort does not register the dose regimen, but it is a relevant idea for the future. We have included this in the methods section and in the discussion as a limitation of the study, as per the suggestion.

COMMENT. Did you have data on lung hyperinflation, IC or Sawr? This point could be important in order to understand the role of a single LABD use in studied COPD population where 89.2% were GOLD 1 & 2, so mild or moderate pts! Why a so good effect? Different evidences underline as a single LABD well work on lung deflation and consequently on dyspnoae (your data reported MRC from 0 to 2 = 99.2% of COPD pts with only a single LABD); see 1) Comparison of the acute effect of tiotropium versus a combination therapy with single inhaler budesonide/formoterol on the degree of resting pulmonary hyperinflation. Respir Med. 2006 Jul;100(7):1277-81. 2)Assessment of acute bronchodilator effects from specific airway resistance changes in stable COPD patients. Respir Physiol Neurobiol. 2014 Jun 15;197:36-45. Furthermore, the bronchodilatation and lung deflation working on dyspnoea and this could have influence also on the treatment satisfaction as well as on the adherence (see Satisfaction with chronic obstructive pulmonary disease treatment: results from a multicenter, observational study. Ther Adv Respir Dis. 2019 Jan-Dec;13:1753466619888128).

ANSWER: We agree with this very interesting and pertinent comment, and would like to thank for allowing us to clarify. The point raised by the reviewer is key to understand how LABD improve symptoms perception in COPD. However, one of the key aspects of TRACE is to be able to apply its findings for clinical decision in daily clinical practice. That is why TRACE does not use diagnostic tools that are simple enough to be available in a routine medical visit for respiratory patients including primary care. Therefore, the TRACE cohort does not collect data from more advanced evaluation techniques like lung volume. Interestingly, FVC was not kept in the final model. Although this could be because of the real-world patient type included. We have added a comment as a limitation in the discussion as per the suggestion.

COMMENT. In general the Discussion is interesting but must be shortened for every point and implemented with a brief point relating to the two previous suggestions.

ANSWER: We have added this comment in the discussion, as per the suggestion.